# Assessment of Ecosystem Services of Wetlands of the Volga–Akhtuba Floodplain

Alexander I. Belyaev, Anna M. Pugacheva and Evgenia A. Korneeva *

Federal Scientific Center of Agroecology, Complex Melioration and Protective Afforestation, Russian Academy of Sciences, University Ave, 97, 400062 Volgograd, Russia
* Correspondence: korneeva.eva@list.ru; Tel.: +7-9178407904

**Abstract:** The economic meaning of measures to water wetlands based on calculations of the economic value of their ecosystem goods and services is insufficiently studied in Russia. In this regard, it is difficult for decision-making authorities to adopt these measures as a strategy for sustainable management of natural resources. The purpose of the research is a monetary assessment of the regional benefits from ecosystem services of wetlands that the local community of the Lower Volga region will receive in connection with the rehabilitation of the Volga–Akhtuba floodplain. The study presents the magnitude and structure of these ecosystem services. The methodology of their economic assessment is given. It is established that by the period of full restoration of the hydrological regime of the Volga–Akhtuba floodplain (2035), the economic value of provisioning services of its wetlands, taking into account inflation and regional pricing, will be USD 87 ha$^{-1}$ year$^{-1}$, the economic value of cultural services—USD 77 ha$^{-1}$ year$^{-1}$, the economic value of regulation and maintenance services—USD 106 ha$^{-1}$ year$^{-1}$. The data obtained indicate the high importance of wetland irrigation measures for the Lower Volga region and allow us to consider them as a means of improving the quality of the environment and solving social problems of the region by decision-making authorities involved in the sustainable management of its development.

**Keywords:** wetlands; ecosystem services; economic assessment; local community; regional development

## 1. Introduction

The Volga–Akhtuba floodplain is a valley located between the Volga River and one of its branches—the Akhtuba River. Most of the floodplain is covered by reservoirs and is located within the Volgograd region, Russia. The main value of the floodplain is represented by wetlands with unique biodiversity, fertile floodplain lands, highly productive flood meadows, key nesting and resting places for birds, and spawning grounds for fish and other aquatic organisms. The floodplain also plays an important regulatory role in the assimilation of dust in the atmospheric air for urban agglomeration [1].

For the past few years, a critical ecological situation has been observed in the Volga–Akhtuba floodplain. For the majority of wetlands, the unstable hydrological regime is the greatest threat here. The decision-making authorities recognize the danger of draining the floodplain due to the critically low water level [2,3]. The main reason for this state of the natural ecosystem was the commissioning of the Volga hydroelectric power station, Volgograd. Under the influence of artificial regulation of the hydrological regime of the Volga–Akhtuba floodplain, the transformation of the ecological parameters of reservoirs and the loss of many useful properties significant for humans occurred. The displacement in time of the flood, the decrease in its volume and duration, and the violations of the temperature regime sharply reduced the reproduction of ecosystem resources of the floodplain [4].

In order to improve the flooding of the Volga–Akhtuba floodplain and increase its environmental sustainability, a project was developed to artificially fill the dried-up Akhtuba

River [5]. It is planned that the hydraulic structures created within the framework of the project will flood the dried-up part of the wetlands of the Volga–Akhtuba floodplain, reduce their ecological tension, and bring the state of the floodplain to the level of pristine nature. Taking into account the significant investments USD 6.37 million) directed to the implementation of this project, it becomes relevant and extremely important to assess the potential benefits that the region will receive due to these measures in the form of ecosystem services of revived floodplain wetlands. These studies should be based on generally accepted valuation methods that are adequate to global monetary estimates, but they should have their own regional characteristics.

It is noted [6] that global wetlands are also subject to degradation. During the 20th century, their area decreased by almost 70% [7]. Along with natural factors, irrational human activity plays an important role in this [8].

In many countries, the degradation of wetlands and the associated deterioration of ecosystem services can lead to a significant loss of human well-being and biodiversity, as well as negative long-term consequences for the economy and communities. Therefore, the protection and restoration of wetlands are necessary for the future sustainability of the planet [9].

Wetlands perform many functions and provide a variety of ecosystem services that contribute to human well-being [10,11]. Among their main contributions are habitat accessibility, water purification, food supply, recreation, and cultural heritage. Wetlands also play an important role in regulating the global environment, preserving the global hydrological cycle, and ensuring human well-being [12,13].

Economic assessment is a powerful tool for managing global wetland resources, providing a means to measure and compare the various benefits of wetlands [14]. There are economic assessments of these natural ecosystems and their biodiversity in the world scientific literature [15–18].

It is established that in most cases, the method of market prices is used to determine the services for the provision (production) of wetlands, and the method of replacement cost (compensation cost) is used to evaluate the recycling services. Cultural services are evaluated mainly on the basis of identified preferences regarding travel and accommodation cost [19,20].

In Russia, the economy of ecosystem services, especially wetlands, is extremely poorly developed both in economic research and in policy documents [21,22]. The current net values of the economic assessment of the consequences of measures to water drained areas, in particular wetlands, presented in the literature, are few and extremely variable, so it is impossible to make any quantitative generalizations for different natural areas or methods. The only conclusion made based on the analysis of the available economic information is that the floodplain flooding potential is real, and it can be economically beneficial under various circumstances.

For the first time in Russia, the federal project of artificial flooding of the Volga–Akhtuba floodplain was conceptualized in order to prioritize the environmental and social sustainability of the region where the hydraulic structures will be located. In other words, the needs of the local population are at the forefront, and not the commercial success of the project.

So, first of all, the role of the floodplain as a regulator of the purity of the atmospheric air of the urban agglomeration is well known. These are the peculiar lungs of the Volga dry steppes and semi-deserts, a kind of filter for neutralizing the products of man-made human activity. The increase in the effect of dust assimilation will be ensured by preventing the drying of forest stands, in particular oak forests, due to flooding of the root systems of trees on the area of the forest fund available in the floodplain. Regional benefits will also be expressed in the form of solving problems with unsatisfactory water supply and water quality, the implementation of recreational functions of the floodplain area—recreation and health improvement of people. In addition, benefits from aquaculture will be received in the form of income from the extraction of floodplain fish resources and amateur fishing, popular among the local population.

Thus, the purpose of the study was a monetary assessment of the regional benefits from ecosystem services of wetlands that the local community of the Lower Volga region will receive in connection with the rehabilitation of the Volga–Akhtuba floodplain.

## 2. Materials and Methods

### 2.1. Case Study Sites

The object "Volga–Akhtuba floodplain" is located in the south of Russia (Figure 1). It occupies about 800 thousand hectares. Its width in places reaches 35 km. The climate of the floodplain territory is sharply continental, arid with little precipitation (200–220 mm). Spring is usually dry and hot, summer is sultry, arid, and winter is cold. The maximum air temperature in some days of July reaches 42 °C, and the minimum in some periods of January drops to 28 °C. During the period from April to October, an average of 115 mm of precipitation falls in the floodplain [23]. High and stable yields of agricultural crops on the meadow lands of the Volga–Akhtuba floodplain can be obtained only with irrigation.

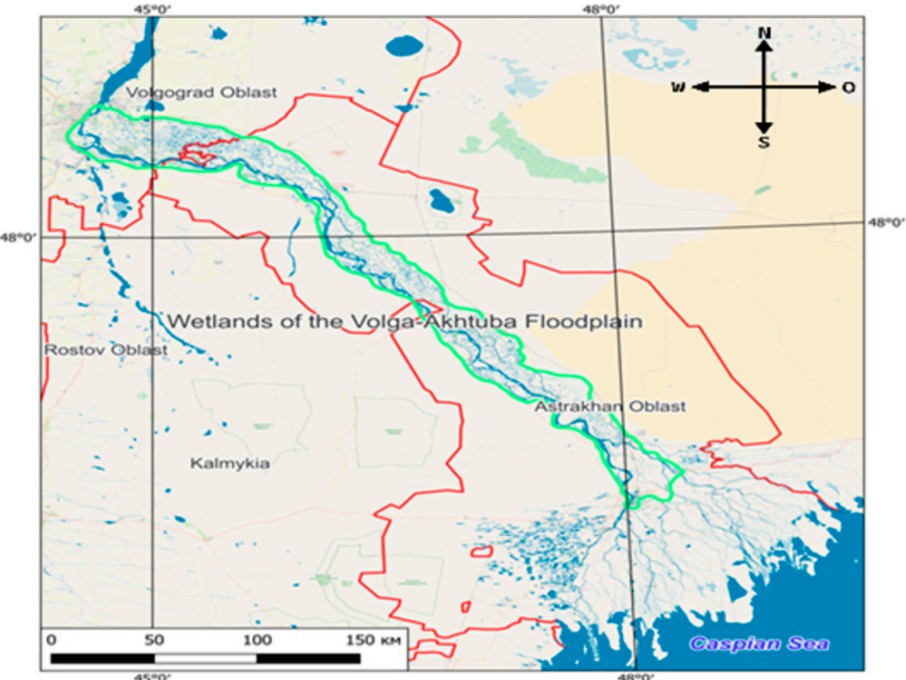

**Figure 1.** Geographical location of the Volga–Akhtuba floodplain.

2.1.1. Assessment of the Ecological State of the Wetlands of the Volga–Akhtuba Floodplain

Before the creation of the Volga hydroelectric power station (1961), the Akhtuba River was the most full-flowing channel of the Volga (Figure 2). It was used for passenger traffic where large passenger ships sailed on it. During the flood, it contributed to the uniform flooding of the Volga–Akhtuba floodplain, which was formed as a single ecological space with pristine nature, flora, and fauna. Flooding of the floodplain territory was favorable because the hollow waters enriched the soil, moistened hayfields, saturated oak forests, and formed vast water areas for spawning and feeding fish. Thanks to the continuous soil-forming process, highly fertile alluvial soils and fresh groundwater were formed here [1].

After the construction of the Volga hydroelectric power station, the natural water flow along the Akhtuba River changed radically [3]. The water level at the beginning of the river decreased by 40–45 cm and became completely dependent on the water level in the downstream of the hydroelectric power station. Once the navigable Akhtuba River became shallow in some areas up to half a meter. Due to the drop in the water level in Akhtuba, the natural hydrological regime worked out by the Volga for centuries has changed radically. All floodplain lakes, rivers, and channels became waterless, and the Volga–Akhtuba floodplain lost its natural watering (Figure 3).

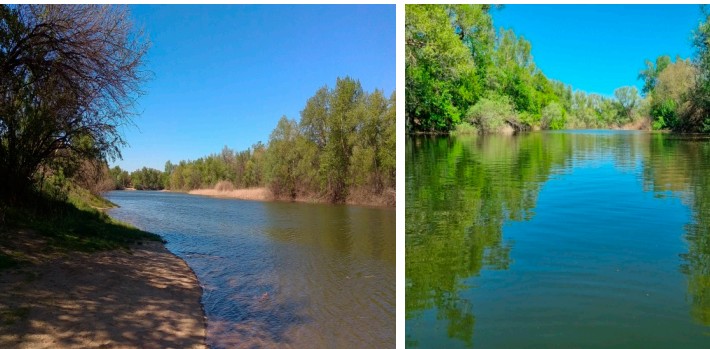

**Figure 2.** The initial state of the Akhtuba River and the Volga–Akhtuba floodplain (Photos of the authors).

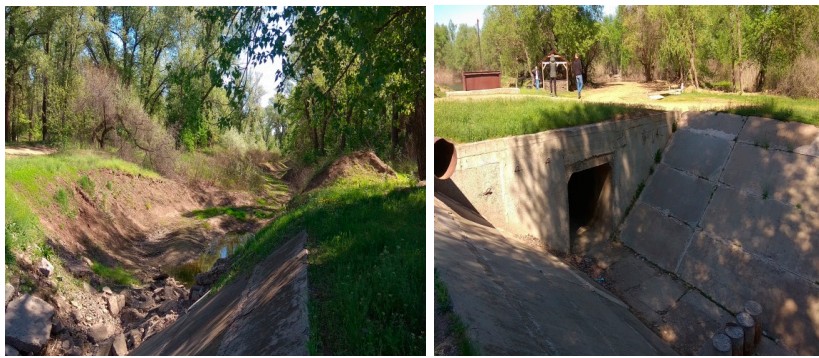

**Figure 3.** The deteriorated state of the Akhtuba River and the Volga–Akhtuba floodplain (Photos of the authors).

With the help of the implementation of the federal project for additional artificial flooding of the Volga–Akhtuba floodplain, it is planned to solve these problems and restore ecological stability and natural balance in the regions of the Lower Volga region. The complex of hydraulic structures under construction will allow creating an artificial hydrological regime on the Akhtuba River and will allow filling the floodplain even in a low-water period (Figure 4).

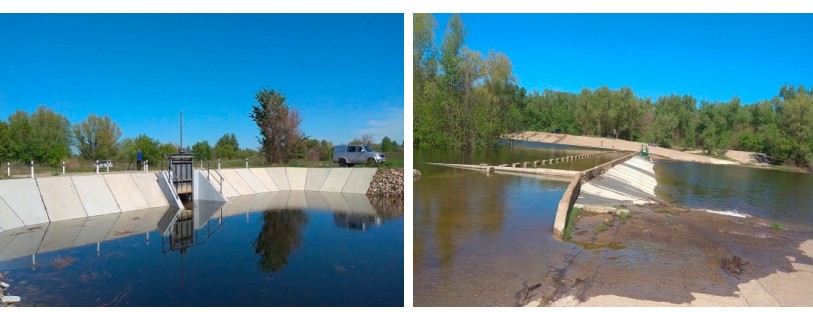

**Figure 4.** Restored condition of the Akhtuba River and the Volga–Akhtuba floodplain (Photos of the authors).

2.1.2. Engineering Parameters of the Project "Additional Flooding of the Akhtuba River and the Volga–Akhtuba Floodplain"

According to the administration of the Volgograd region the complex of projected hydraulic structures consists of the following objects:

1. Pumping station;
2. The hydroelectric power station building, consisting of three units with a total capacity of 31.2 MW;
3. Dam regulators with a crest height of 2.6 m with a maximum water level of −0.7 m;

4. The system of hydraulic structures on large rivers will allow the floodplain to receive the necessary water, regardless of the discharges of the Volga hydroelectric power station;

5. A 32 km long culvert with a possible throughput capacity of 165–1000 m$^3$ s$^{-1}$;

6. Estuary (reservoir) with a mark from $-10$ to $-3$ m, depending on the season.

So, above the Volga hydroelectric power station from the Volga to the Akhtuba, a culvert 32 km long, 12 m deep, and 100 m wide will be dug. Then, the water will pass through a new hydroelectric power plant with a capacity of 31.2 MW. The water will flow into the estuary, which is formed on the Akhtuba River with the help of two dams that will regulate the water level in it. Depending on the time of year, it will be possible to change the mark of the estuary. The flow rate to the estuary is planned from 165 to 1000 cubic meters per second. Its use in the inter-war period will raise the water level in the Akhtuba River by 3 m, and in the flood period by another 4 m. Thus, the gravity flooding of the Volga–Akhtuba floodplain will be ensured, as well as, if necessary, the accumulation of water and its redistribution to the needs of the floodplain during the year [24].

## 2.2. Data Collection

To calculate the regional ecosystem services of wetlands, quantitative biophysical data were used, which were obtained within the framework of the project "Economic assessment and assessment of the impact on the needs of agriculture of additional flooding of the Volga–Akhtuba floodplain", developed by Federal Scientific Center of Agroecology, Complex Melioration and Protective Afforestation, Russian Academy of Sciences.

The structure of the total regional ecosystem product of wetlands included the value expression of water management, recreational, air purification, and aquaculture benefits that will be received by the local community of the region from the watered wetlands of the floodplain. The main biophysical parameters of these benefits were established on the basis of official data of the Committee of Natural Resources, Forestry and Ecology of the Volgograd region, Russia [25].

Quantitative data on the normative water consumption of the population were obtained in the Department of Agriculture and Environment Statistics of the State Statistics Service for the Volgograd region [26].

When calculating recreational ecosystem services of wetlands, the available data on permissible recreational loads on the floodplain territory and aquatic complexes of the Volga–Akhtuba floodplain were used [27].

Data on the air-purifying capacity of forest stands were established on the basis of existing materials [28] by the amount of dust filtered by forest stands per year.

The project catch of fish resources was determined based on the materials of the Committee of Agriculture of the Volgograd region [29].

The established quantitative parameters of these ecosystem services were then converted into monetary equivalent using generally accepted methods of economic assessment [30–35].

## 2.3. Data Analyses

Methods Used for the Economic Assessment of Ecosystem Services Provided by Restored Wetlands

Wetlands provide a variety of goods and services. In any evaluation study, it is important to identify all the benefits that are relevant to this wetland under study. We started from the provisions of the Concept of Total Economic Value (TEV), which underlies the assessment of ecosystem benefits [36]. Thus, from an economic point of view, ecosystem services correspond to various elements of TEV, including direct use values (provisioning and cultural services) and indirect use values (regulation and maintenance), according to CICES categorisation [37].

The evaluation study uses various data collection methods and information sources, including both primary and secondary data collection. In our research, secondary data collection was used. It included a literary review of documents and state reports containing

information about the wetlands under study [25,26,29], as well as scientific articles and publications [27,28] (Table 1).

**Table 1.** Methodology of economic assessment of ecosystem services of wetlands based on the entire watered area of the Volga–Akhtuba floodplain.

| Ecosystem Service | Value | Evaluation Method | Biophysical Indicator | Reference of Biophysical Indicators | Unit Price (2021: Price in USD Calculated Using PPP [38]) | Reference of Unit Prices |
|---|---|---|---|---|---|---|
| Provisioning: water used for drinking and other household use | Direct use Value | Market prices: cold water | 375 thousand m³ of water | [25,26] | USD 0.69 per m³ | [39] |
| Provisioning: fish caught | Direct use Value | Market price: fish | 144 tons of fish | [27] | USD 3.66 per kg | Local market |
| Cultural: Recreational | Direct use Value | Market price: stay fee | 300 thousand people | [27] | USD 2.32 per user | [40] |
| Regulation and maintenance: Water purification | Indirect use Value | Replacement cost: water treatment | 375 thousand m³ of water | [25,26] | USD 0.92 per m³ | [39] |
| Regulation and maintenance: Air purification | Indirect use Value | Replacement cost: air treatment | 0.85 millions of tons of dust | [28] | USD 0.73 per ton | [41,42] |

Benefits of ecosystem service (BES, USD year$^{-1}$) include provisioning services (PS), cultural services (CS), and regulation and maintenance services (RMS):

$$BES = PS + CS + RMS \tag{1}$$

Provisioning services include water for drinking and other household use ($W_{drink}$) и fish caught ($F_{ish}$):

$$PS = W_{drink} + F_{ish} \tag{2}$$

These indicators are defined as follows:

$$W_{drink} = \sum_{i=1}^{n} Q_{water} \times Pr1_i \tag{3}$$

In the formula, $Q_{water}$—the amount of water consumption for household needs of the population living on the territory of the Volga–Akhtuba floodplain, m³ year$^{-1}$, $Pr1_i$—the price of the water in the region, USD.

$$F_{ish} = \sum_{i=1}^{n} Q_{fish} \times Pr2_i \tag{4}$$

In the formula, $Q_{fish}$—project catch of fish, tons year$^{-1}$; $Pr2_i$—the price of fish on the regional market, USD.

Cultural services are evaluated based on the permissible recreational capacity of the Volga–Akhtuba floodplain (without damage to the environment and natural ecosystems) according to the following formula:

$$CS = \sum_{i=1}^{n} RC \times p_i \tag{5}$$

In the formula, $RC$—recreational capacity (vacationers), users year$^{-1}$; $p_i$—the fee for staying in nature parks, USD.

Regulation and maintenance services include water purification ($W_{pur}$) и clean air regulation ($A_{ir}$):

$$RMS = W_{pur} + A_{ir} \tag{6}$$

The value of these indicators is calculated as follows:

$$W_{pur} = \sum_{i=1}^{n} Q_{water} \times C1_i \tag{7}$$

In the formula, $Q_{water}$—the amount of water consumption for household needs of the population living on the territory of the Volga–Akhtuba floodplain, $m^3$ year$^{-1}$; $C1_i$—the price of water purification, USD.

$$A_{ir} = \sum_{i=1}^{n} Q_{dust} \times C2_i \tag{8}$$

In the formula, $Q_{dust}$—amount of dust filtered by forest plantations (on average for hardwoods with an average current amount of deposited dust of 1.96 g per $m^2$ of foliage or 450.8 g per tree in the period between rains [28]), tons year$^{-1}$; $C2_i$—the price of capturing and neutralizing industrial dust (suspended solids 2.5 RM), which includes all the actual costs of operating enterprises for the protection of atmospheric air (operation of dust-collecting equipment), USD.

The paper takes into account the future inflation rates (projected consumer price indices on average for the year), established on the basis of official data of the Ministry of Economic Development of the Russian Federation [43].

We standardized values to USD ha$^{-1}$ year$^{-1}$ in 2021 prices using purchasing power parity adjusted exchange rate [38]. The forecast values of the economic value of ecosystem services are obtained taking into account inflation for each year, as well as using the PPP of 2021 for future years, for example, for 2035. This is based on the assumption that the pricing system for the basic set of goods and services will not change dramatically in the country.

Unit values (ha$^{-1}$ year$^{-1}$) is obtained by dividing the annual economic value of ecosystem services by the watered area, which changes annually as the system develops.

To calculate the economic value for 2007, the local currency values for 2007 were first calculated using the inflation rate. Then, the 2007 PPP was used to calculate the values in US dollars.

### 2.4. Caveats of Study

The economic analysis takes into account the macroeconomic perspective of the federal project "Additional flooding of the Volga–Akhtuba floodplain" and assesses its benefits from the perspective of the economy and society. When calculating, it is assumed that from 2026 (the year of commissioning of the complex of hydraulic structures providing water supply to the Akhtuba River), the process of restoring the wetlands of the Volga–Akhtuba floodplain begins. In the year when the project reaches full capacity (2035), the economic effect ends, that is, at the moment of complete restoration of wetlands with key places of floodplain oak forests, flood meadows, and fish spawning grounds. Thus, the time horizon of the study is 10 years. Note that this time horizon characterizes only the prospect of project implementation. It is obvious that ecosystem services will continue to be provided by wetlands in the long term. In addition, the structure of the total ecosystem product also includes other market and non-market ecosystem services. Consequently, the presented indicators reflect only the minimum values.

### 3. Results

At the economic assessment of ecosystem services provided by restored wetlands, it was assumed that the hydraulic structures would not work at full capacity immediately, but after a certain time. In order to fill and properly maintain the hydrological regime throughout the Volga–Akhtuba floodplain during the year, it is necessary not only to

create hydraulic engineering facilities, but also to debug them, as well as clearing existing dried-up reservoirs and cleaning up the littered territory.

Thus, the project provides that the technical operation of hydraulic facilities will allow water to be supplied to the Akhtuba River and watered and cleaned on average only 10% of the floodplain area per year, starting from 2026. It is planned that by 2035 the work on the creation and debugging of hydraulic structures will be completed and they will operate at full 100% capacity. We assume that the biophysical indicators of ecosystem services, mainly related to water, will increase in a similar proportion as their economic estimates.

The number of vacationers on the territory of the Volga–Akhtuba floodplain is regulated by scientifically-based norms of recreational loads.

As for ecosystem services for cleaning the air basin from dust, the trees located on the territory of the floodplain wetlands provide these services at the present time. It was assumed that the flooding of the Volga–Akhtuba floodplain would provide the necessary hydrological conditions to prevent the drying of existing forest stands, due to flooding of their root systems. Therefore, biophysical indicators of these ecosystem services are fully taken into account in the calculations starting from 2026. Further, the inflation coefficients are used in the calculations.

It should be noted that by 2035, the project plans to reach an ultra-stable level of supply of all ecosystem services of wetlands (the level of their initial state) in order to fully satisfy the existing demand from the local population for the services of the Volga–Akhtuba floodplain.

Calculations show (Table 2) that the standard of water supply to the population (375 thousand m$^3$) can be brought to 100% only in 2035, since there will be a gradual filling of dried-up reservoirs located on the territory of the floodplain. Thus, 10% of the standard water supply (37.5 thousand m$^3$) will be available for drinking water per year. This also applies to the project catch of fish (14.4 tons per year). The total benefits from provisioning services by the period 2035 will amount to USD 1.36 million (adjusted for inflation), including the economic value of water for drinking and household use (USD 448,071), and the economic value of fish (USD 912,663).

The calculation of the economic value of cultural services based on the regulatory recreational load on the floodplain ecosystems of the Volga–Akhtuba floodplain shows that by 2035 its value will be equal to USD 1.2 million. This will be provided with the possibility of the annual presence of 300 thousand vacationers. Currently, staying in the floodplain is regulated by the number of only 30 thousand people per year.

The estimated economic value of regulatory and maintenance services for the period of full restoration of the water supply of the Akhtuba River, taking into account inflation, will be USD 1.2 million. Of this amount, the ecosystem service water purification will account for 36% (USD 597,428).

The assessment of ecosystem services for the assimilation of dust by trees shows that the economic value of these services for the period of the beginning of the construction of hydraulic structures (2026) will amount to USD 754,933 per forest fund (deciduous forest-forming species) located within the wetlands of the Volga–Akhtuba floodplain. By 2035, the economic value of this service, adjusted for inflation, will be USD 1.1 million. It should be noted that the assessment of the economic value of ecosystem services for air quality regulation was made with simplification—dust is absorbed by the leaves of trees that have reached middle age (20–40 years). At the same time, the average value of dust absorption for hardwoods common in the Volga–Akhtuba floodplain (oak, poplar, aspen, ash) was taken into account.

In general, it can be argued that for the period of completion of the construction of hydraulic structures in the Volga–Akhtuba floodplain (2035), the restored wetlands will generate the economic value of their ecosystem services in the amount of USD 4.3 million per 15,704 hectares of wetlands.

Based on 1 hectare of these lands, services worth USD 270 will be provided by 2035 (adjusted for inflation) (Table 3). The present value of these ecosystem services (in prices of 2021) is equal to USD 156 ha$^{-1}$ year$^{-1}$. Of this, USD 50 ha$^{-1}$ year$^{-1}$ is the economic value

of provisioning services, USD 44 ha$^{-1}$ year$^{-1}$ is the economic value of cultural services, and USD 62 ha$^{-1}$ year$^{-1}$ is the economic value of regulation and maintenance services.

**Table 2.** Economic assessment of ecosystem services provided by restored wetlands of the Volga–Akhtuba floodplain (based on 15,704 hectares of wetlands).

| Calculated Indicators | Floodplain Flooding Level, % of the Total Area | | | | |
|---|---|---|---|---|---|
| | **10** | **40** | **60** | **80** | **100** |
| | **2026** | **2029** | **2031** | **2033** | **2035** |
| | **Start of Construction of Hydraulic Structures** | **Debugging of Hydraulic Structures, Clearing of Territory and Reservoirs** | | | **Full Operation of Hydraulic Structures** |
| Provisioning services: | | | | | |
| The standard of water consumption for household needs of the population, thousand (m$^3$/year) | 37.5 | 150.0 | 225.0 | 300.0 | 375.0 |
| Projected/estimated catch of fish (tons year$^{-1}$) | 14.4 | 57.6 | 86.4 | 115.2 | 144.0 |
| Economic value of water for drinking and household use (USD year$^{-1}$) | 31,481 | 141,647 | 229,808 | 331,414 | 448,071 |
| Economic value of fish (USD year$^{-1}$) | 64,122 | 288,516 | 468,089 | 675,046 | 912,663 |
| Cultural services: | | | | | |
| Recreational capacity of wetlands (thousand people/year) | 30 | 120 | 180 | 240 | 300 |
| Economic value of recreation (USD year$^{-1}$) | 84,679 | 381,010 | 618,150 | 891,455 | 1,205,247 |
| Regulation and maintenance services: | | | | | |
| The standard of water consumption for household needs of the population (thousand m$^3$/year) | 37.5 | 150.0 | 225.0 | 300.0 | 375.0 |
| The amount of filtered dust (million tons/year) | 0.85 | 0.85 | 0.85 | 0.85 | 0.85 |
| Economic value of water purification (USD year$^{-1}$) | 41,975 | 188,863 | 306,411 | 441,885 | 597,428 |
| Economic value of air purification (USD year$^{-1}$) | 754,933 | 849,197 | 918,492 | 993,440 | 1,074,505 |
| The economic value of ecosystem services for the year when the system is full operating (USD year$^{-1}$) | - | - | - | - | 4.238 million |

**Table 3.** Economic assessment of ecosystem services, which changes as the system develops (USD ha$^{-1}$ year$^{-1}$).

| Economic Value of Ecosystem Services | 2007 (The Initial Level) | 2026 | 2029 | 2031 | 2033 | 2035 (The Forecast Level) |
|---|---|---|---|---|---|---|
| Area of restored wetlands, ha | - | 1570 | 6282 | 9422 | 12,563 | 15,704 |
| Economic value of provisioning services | 35 | 61 | 68 | 74 | 80 | 87 |
| Economic value of cultural services | 31 | 54 | 61 | 66 | 71 | 77 |
| Economic value of regulation and maintenance services | 43 | 75 | 84 | 91 | 98 | 106 |
| Total economic value | 109 | 190 | 213 | 231 | 249 | 270 |

Thus, calculations of the economic value of ecosystem services of wetlands, which is based on actual use (estimated in the future), indicate that, on the one hand, this value will increase due to the expansion of the watered area of wetlands as the system develops. On the other hand, the economic value of their ecosystem services will increase as a result of the development of inflationary processes in the country.

## 4. Discussion

Wetlands are one of the most biologically productive ecosystems. They play an important role in the water cycle, regulating its flows and supporting life. The ecosystem services of wetlands are far superior to those of terrestrial ecosystems. Among other things, they provide essential food supplies, including freshwater fish and fresh water [44].

Quantification of ecosystem services is important for policy and management decisions, such as protection and restoration of coastal wetlands [45]. However, the biggest problem for assessing ecosystem services is the lack of knowledge to link changes in the structure and functions of ecosystems with the production of valuable goods and services. Ecosystem services of wetlands have many characteristics of public goods and are not always realized in economic markets (for example, habitat for biodiversity, water quality) [46,47]. In this regard, it is impossible to fully establish a link between the functioning of ecosystems and goods and services that are produced for society [48]. As a result, it is often difficult to determine the monetary value of many wetland ecosystem services that could be used to develop wetland conservation policies.

It is noted that in current studies there are no uniform indicators for assessing ecosystem services of wetlands [49]. Most existing studies double-count the economic value that ecosystems provide to beneficiaries [50]. Difficulties in addressing the environmental and social aspects of ecosystem services also hamper policy makers with regard to sustainable wetland management [51].

De Groot et al. [52], based on the study of 168 estimates, established maximum and minimum estimates for inland wetlands. In 2007 prices, they are, respectively, USD 104,924 ha$^{-1}$ year$^{-1}$ and USD 3018 ha$^{-1}$ year$^{-1}$. Total of median values is USD 16,534 ha$^{-1}$ year$^{-1}$. Note that it is a global estimate, which is calculated as a mean of available unit values that time, and the calculation methods of the original studies are mixed. Additionally, de Groot et al. estimated the value of ecosystem services in monetary units provided by ten major biomes, including inland wetlands. The values were expressed in terms of USD 2007 ha$^{-1}$ year$^{-1}$ on the basis of PPP. Provisioning services of inland wetlands estimated in size USD 1659 ha$^{-1}$ year$^{-1}$, including food (USD 614 ha$^{-1}$ year$^{-1}$) and water (USD 408 ha$^{-1}$ year$^{-1}$). Regulating services are USD 17,364 ha$^{-1}$ year$^{-1}$. The economic value of air quality regulation for inland wetlands is not presented, however, for tropical forests it is USD 12 ha$^{-1}$ year$^{-1}$. Cultural services are estimated at USD 4203 ha$^{-1}$ year$^{-1}$, including the economic value of recreation is USD2211 ha$^{-1}$ year$^{-1}$.

In a study by Brander et al. [53] presented a meta-analysis of the economic valuation literature on ecosystem services provided by wetlands in agricultural landscapes. The data were obtained in 2007 prices using the adjusted PPP exchange rate. The main valuation method is the value transfer using a value function estimated through a meta-analysis of the results of multiple primary studies. The mean (median) values are found to be USD 6923 (427) ha$^{-1}$ year$^{-1}$ for flood control; USD 3389 (57) ha$^{-1}$ year$^{-1}$ for water supply; and USD 5788 (243) ha$^{-1}$ year$^{-1}$ for water quality (in 2007 prices). It is noted that low average specific values are observed in countries with sparsely populated agricultural areas and relatively abundant wetlands (for example, Canada and Russia) and high values in countries with densely populated and relatively few wetlands (for example, Japan). So, mean unit value for wetland regulating services in agricultural landscapes in the USA is USD 1490 ha$^{-1}$ year$^{-1}$; in Canada—$223 ha$^{-1}$ year$^{-1}$; in Western Europe—$2353 ha$^{-1}$ year$^{-1}$; in Central Europe—$1743 ha$^{-1}$ year$^{-1}$; in China—$1502 ha$^{-1}$ year$^{-1}$; in Russia—$297 ha$^{-1}$ year$^{-1}$; in Japan—$5817 ha$^{-1}$ year$^{-1}$.

Generalized global data on the monetary value of services provided by inland wetlands of the world are in the range of USD 981–44,597 ha$^{-1}$ year$^{-1}$ (2007 values). Of these, USD 2–9709 ha$^{-1}$ year$^{-1}$ accounts for provisioning services, USD 321–23,018 ha$^{-1}$ year$^{-1}$ accounts for regulating services, USD 648–8399 ha$^{-1}$ year$^{-1}$ for cultural services of wetlands [16]. The assessment of the economic value of these services was carried out by the method of market prices, the economic value of prevented environmental damage, the replacement cost, and other methods.

The wetlands of the Volga–Akhtuba floodplain belong to the type of inland wetlands that were included in the above range of assessments. If we present the values of the economic value of ecosystem services of wetlands obtained by us at the price level of 2007 (using official data on inflation in the country [42] and the PPP level for 2007 [38]), we will realise that our estimates of economic value of regulation and maintenance and culture services are somewhat less than the data presented, which is explained by a small number of assessed ecosystem services of wetlands, as well as the peculiarities of regional pricing and commodity-monetary policy of the country that significantly differ from the world ones. In addition, the reason for this is the underdevelopment of the national market of ecosystem services. The largest amount (USD 43 ha$^{-1}$ year$^{-1}$) falls on regulation and maintenance services. The lowest amount is for cultural services (USD 31 ha$^{-1}$ year$^{-1}$). The economic value of provisioning services is adequate to the global estimates presented and amounts to USD 35 ha$^{-1}$ year$^{-1}$. The total economic cost of ecosystem services is USD 109 per 1 ha per year in 2007 prices.

Despite the low estimated values of the economic value of the restored wetlands, they are nevertheless very valuable. A good reference equivalent for the value of wetlands can be the value of a hectare of forest land. Thus, in the studied region, the average value of forest lands in 2007 prices [54], estimated using PPP level for 2007 [38], equal to USD 1065 per 1 ha.

In general, the available wetland assessment studies are diverse in terms of the values obtained, the wetlands assessed, and the characteristics of the studies. These differences are mainly due to different assessment methodology. It should be noted that the assessment of ecosystem services is based on the calculation method—willingness to pay exceeds the available estimates based on the methodology of market prices. This indicates a high demand on the part of the population for the services of natural ecosystems, in particular wetlands.

However, the most of the world's environmental strategies aimed at protecting and improving wetlands focus on public interests not only solely for the purpose of improving the ecological and social well-being of regions, but also their economic interests [55].

Currently, the restoration of wetlands is a priority direction of development in many countries. Ensuring the conservation and increasing the benefits of wetlands is recognized as a key element of the transition to a "green economy". Investments in the conservation and restoration of ecosystem services of wetlands are a valuable tool for policy decisions [56]. These investments are not a burden on the economies of countries and bring benefits.

So, in the USA (Catskill Mountains), a natural capital solution worth USD 2 billion (restoration and maintenance of the watershed) versus a technological solution worth USD 7 billion (pretreatment installation) made it possible to prevent an increase in water tariffs for the population [57].

An economic study in the framework of TEEB [58] showed that the economic value of livelihood derived from degraded wetlands is only 34 percent of what can be achieved after investments in ecosystem restoration. At the same time, the total economic value (EUR 182,000 or USD 178,360) of the restored wetlands is more than twice the economic value of restoration work (EUR 86,000 or USD 84,280), which indicates a decent return on investment.

We also believe that the restoration of wetlands is an expedient and cost-effective measure.

In the absence of local wetland assessment studies, our data can be used to compare the value of wetlands in landscapes with similar natural and climatic conditions. Our assessment tools can also be useful in land use planning and wetland conservation policy development.

We have made an attempt to estimate in the first approximation some obvious potential of ecosystem services of wetlands in Russia. These assessments will play an important role in justifying the feasibility of investments in the creation of a complex of hydraulic structures for watering drained floodplain lands. We have presented our interpretation of estimates based on modern research methods, but conditioned by regional peculiarities of pricing and tariff policy in the country. The estimated values of ecosystem services obtained indicate significant benefits of the project for the local population. Given that ecosystem services are becoming more and more in demand in the country, we can expect that their value will only increase in the future.

## 5. Conclusions

The article attempts to evaluate regional ecosystem services based on available methods in the context of a wetlands restoration project. It contains proposals on recommended indicators and criteria for assessing ecosystem services as a basis for decision-making on wetland restoration projects in Russia. Given the huge uncertainties associated with this, it is not possible to obtain a very accurate forecast of the future value of ecosystem services. At the same time, the values of ecosystem services presented here are spatially and temporally specific. They give an idea of the economic aspects that previously went unnoticed, and allow us to point out the high expediency of investing in an ecosystem service, since the benefits from its provision pay off the economic value several times and ensure the sustainable development of local communities for a long time.

The assessment of ecosystem services is an important tool for justifying decision-making on a specific wetland restoration site. Using the tools of this assessment gives decision makers and other stakeholders the opportunity to better understand how wetland ecosystems improve the quality of life and well-being of people. It is important to recognize that the values achieved are initial and demonstrate the need for a large number of additional studies. This information is necessary, at least, for an objective assessment of the importance of taking into account ecosystem services in the justification of environmental protection measures in the country in modern economic conditions.

**Author Contributions:** A.I.B., Data curation; A.M.P., conceptualization; formal analysis, methodology, software, E.A.K. All authors have read and agreed to the published version of the manuscript.

**Funding:** The article has been prepared in accordance with the state task of the Russian Ministry of Education and Science No. FNFE-2022-0015 to Federal Scientific Center of Agro-ecology, Complex Melioration and Protective Afforestation Russian Academy of Sciences.

**Institutional Review Board Statement:** Not applicable.

**Informed Consent Statement:** Not applicable.

**Data Availability Statement:** Data available on request.

**Conflicts of Interest:** The authors declare no conflict of interest.

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
