# Peer review of "Assessment of Ecosystem Services of Wetlands of the Volga–Akhtuba Floodplain"

_sustainability, doi:10.3390/su141811240_

Round 1
Reviewer 1 Report
In this paper, Alexander I. Belyaev et al examined ecosystem services of wetlands of the Volga-Akhtuba floodplain. This paper is well-written with an interesting idea and clear result. It is also important information or recommendation for policymakers or scientists. I recommend this paper be published after some modifications. Below are my viewpoints:
Major points:
(1) The quality of visualization can be considered. For example, the authors may consider adding some figures or TOC Art or map to improve the readers' understanding of the paper.
(2) Please make sure that the estimation of ecosystem service is accurate. For example, Line 243 "the effect of the aquaculture supply by the time of commissioning of hydraulic structures at full capacity will amount to EUR 1.811 million in terms of fish products"
(3) In conclusion, the authors stated that in Line 340 "it is not possible to obtain a very accurate assessment of the value of ecosystem services." Is that possible for the authors to make the recommendation: to invest or not to invest in the ecosystem service? Please also briefly show the reason(s).
Minor points:
(1) Line 112 "hectares" Line 243 "ha" please make the unit consistent if possible
(2) Line 306 to Line 308 ", the average unit value of wetlands in the USA is $ 1,490 ha-1 year-1, in Canada – $ 223, in Western Europe – $2,353, in Central Europe – $1,743, in China –$1,502, in Russia – $297. " Why are the numbers in USA and Canada so different? Please check the number.
(3) Line 275 Discussion: The authors may link the investment in the ecosystem to the total finance situation of the countries, showing it will not take any burdens on the country and it will provide benefits.
Reviewer 2 Report
Based on some research methods (system analysis, compensation cost, use fee, discharge fee, fishing quota, recreational capacity, market price), this paper evaluates the ecosystem service function of the Volga ahtuba floodplain wetland in Russia, aiming to improve the environmental quality and solve the social problems in the region through the decision-making authorities participating in the sustainable development management. This article has certain value and practical significance. However, there is a certain gap between the article and this journal in terms of writing structure, logical thinking and writing methods. The author is expected to carefully revise and enrich the full text. The specific opinions are as follows:
1. The author is requested to rewrite the abstract on the basis of revising the full text. It is suggested to write it from the aspects of research significance, research purpose, research methods, conclusions and suggestions.
2. In the introduction part, it is suggested that the author should modify the original manuscript, mainly summarize the research progress of scholars at home and abroad, put forward deficiencies and defects, and finally describe the significance and purpose of the regional research in this paper. It can supplement the advantages or disadvantages of this research method.
3. The discussion part needs to be rewritten. It is written in chapters from the significance of wetland ecosystem service assessment, the comparison between this research method and the current research method, and the analysis of the research results.
4. The conclusion part needs to supplement the future outlook or corresponding countermeasures and suggestions (a few words)
5. It is suggested to supplement the overview map of the study area and give a brief introduction
6. There are also other details in this article, such as 2.2.
7. Please check the author's full text for specific statement problems
8. Is the research method innovative? I can't see it right now.
Reviewer 3 Report
Introduction: It provides sufficient details, but it should be more focused on your research objectives. As there is no separate literature survey, it would be beneficial to include some of the previous more relevant studies.
Material and Method: The majority of the required details are not clearly explained here. I would suggest including a table that explains all of the variables that the authors used to perform the estimations/calculations. Variable definitions, data collection methods for each variable, and estimation procedures for all parameters must be explained in more detail.
According to Equation 1 the ‘economic cost of ecosystem provision of wetlands for ecological and social sustainability of the region includes the provision of services for water management (Ewater), recreational (cultural) resource (Еrec), air purity regulation (Edust) and aquaculture supply (Eaqua) of ecosystems in the region annually’. Equations 2,3,4,5 explains each component but with less information. For example measurement units of each component, empirical procedure of estimation..etc. are not clear. Therefore, it is required to explain them more clearly.
Results and discussion: Authors can include a table (first table) that contains descriptive statistics or key parameter values for the main variables.
Discussion and conclusions should be more focused. Any policies that may be drowned out by the finding should be included.
Round 2
Reviewer 2 Report
Agree to publish
